# Progesterone and Estrogen Signaling in the Endometrium: What Goes Wrong in Endometriosis?

**DOI:** 10.3390/ijms20153822

**Published:** 2019-08-05

**Authors:** Ryan M. Marquardt, Tae Hoon Kim, Jung-Ho Shin, Jae-Wook Jeong

**Affiliations:** 1Department of Obstetrics, Gynecology & Reproductive Biology, Michigan State University, Grand Rapids, MI 49503, USA; 2Cell and Molecular Biology Program, Michigan State University, East Lansing, MI 48824, USA; 3Division of Reproductive Endocrinology, Department of Obstetrics and Gynecology, Guro Hospital, Korea University Medical Center, Seoul 08318, Korea

**Keywords:** progesterone, estrogen, endometrium, infertility, endometriosis, progesterone resistance

## Abstract

In the healthy endometrium, progesterone and estrogen signaling coordinate in a tightly regulated, dynamic interplay to drive a normal menstrual cycle and promote an embryo-receptive state to allow implantation during the window of receptivity. It is well-established that progesterone and estrogen act primarily through their cognate receptors to set off cascades of signaling pathways and enact large-scale gene expression programs. In endometriosis, when endometrial tissue grows outside the uterine cavity, progesterone and estrogen signaling are disrupted, commonly resulting in progesterone resistance and estrogen dominance. This hormone imbalance leads to heightened inflammation and may also increase the pelvic pain of the disease and decrease endometrial receptivity to embryo implantation. This review focuses on the molecular mechanisms governing progesterone and estrogen signaling supporting endometrial function and how they become dysregulated in endometriosis. Understanding how these mechanisms contribute to the pelvic pain and infertility associated with endometriosis will open new avenues of targeted medical therapies to give relief to the millions of women suffering its effects.

## 1. Introduction

The endometrium is a complex and dynamic tissue composed of epithelial cells, both luminal and glandular, surrounded by supporting stromal cells, together comprising the innermost layer of the uterus. The primary function of the uterus is supporting fertility, and the endometrium is the layer critically involved in receiving an embryo, facilitating implantation and decidualization, and supporting embryo growth and development until placentation. Successful pregnancy establishment requires an endometrium that is receptive to blastocyst invasion and ready to undergo decidualization, which is dependent upon hormonally regulated molecular processes that allow pregnancy establishment during the period of the menstrual cycle known as the window of receptivity [1,2]. The progesterone (P4) and estrogen (E2)-responsive signaling pathways integral for early pregnancy success are primarily induced through their cognate nuclear receptors, the progesterone receptor (PGR) and estrogen receptors (ESR1 and ESR2), respectively. These pathways are regulated in an epithelial and stromal compartment-specific manner in the endometrium [3,4,5]. E2 induces epithelial proliferation to build endometrial thickness during the proliferative phase of the menstrual cycle, then P4 inhibits E2-induced proliferation and allows stromal cells to begin decidualization during the secretory phase. When the tightly regulated balance of epithelial-stromal P4 and E2 signaling is lost, P4 resistance and E2 dominance are prone to ensue, potentially leading to uterine diseases such as endometriosis [6,7].

Endometriosis is a common uterine disease characterized by the growth of endometrium-like tissue outside the uterine cavity [8,9]. Approximately 10% of reproductive-aged women suffer from this condition, which is often accompanied by chronic pain and infertility. Unfortunately, the etiology of endometriosis is not sufficiently understood to enable consistently effective treatment options. However, it is clear that functional dysregulation of the ovarian steroid hormones P4 and E2 and their downstream signaling targets plays an important role in the development and maintenance of the disease as well as its effects on the eutopic endometrium, primarily through E2-driven inflammation and P4 resistance [6,7].

This review will cover the known roles of P4 and E2 signaling in maintaining endometrial homeostasis and supporting pregnancy establishment before turning to focus on the mechanisms of the P4 and E2 signaling dysregulation of endometriosis, how this dysregulation impacts the clinical symptoms of endometriosis, and how hormone treatment strategies attempt to correct it.

## 2. Steroid Hormone Regulation of Endometrial Function

Studies in mice have been critical to understanding the functions of P4 and E2 in the mammalian uterus during early pregnancy [1,2,10,11]. Compared with the lengthy human menstrual cycle (28–30 days), mice undergo a short estrous cycle (4–5 days), but the receptive window of both species is regulated in a parallel manner by P4 and E2. In mice, a mating event, defined as gestation day (GD) 0, sets off a cascade of hormone signaling events, beginning with a preovulatory E2 surge from GD 0.5–1.5 to induce epithelial proliferation [2]. By GD 2.5, increased P4 secretions from the corpus luteum dominate, promoting stromal proliferation and inhibiting E2-induced epithelial proliferation. Next, a nidatory E2 surge on GD 3.5 acts in concert with P4 regulation to prepare the receptive endometrium on GD 4–5. The invading blastocyst then induces a decidualization reaction of the P4-primed stromal cells, where they differentiate into morphologically and functionally unique cells to surround the implanting embryo and support growth until placentation, all under critical continued P4 regulation. The primary mediators of these P4 and E2-induced events are their cognate nuclear receptors, transcriptional coregulators, and downstream signaling targets.

### 2.1. Progesterone Receptors and Progesterone Signaling

The basic endometrial function of PGR has been known for some time and recently comprehensively reviewed elsewhere [2,11,12,13,14], so this discussion will briefly summarize relevant details while focusing on recent findings of functionally relevant PGR signaling regulators and downstream mediators. PGR expression is induced by E2 action through ESR1, and in turn PGR inhibits ESR1 expression, creating a fine-tuned feedback system to balance downstream effects [12]. PGR is expressed as primarily two functionally distinct isoforms, PR-A and PR-B, transcribed from two promoters in the same gene, resulting in the PR-A protein being 164 amino acids shorter than PR-B [15]. Null-mutation of both isoforms (PRKO) caused sterility in the female mouse due to numerous reproductive abnormalities, including severely reduced or absent ovulation, uterine hyperplasia and a lack of decidualization response, severely limited mammary gland development, and an inability to exhibit sexual behavior [16]. Specific deletion of PR-A [17] or PR-B [18] showed that PR-A is the primary driver of uterine PGR function and is sufficient for fertility, while PR-B is critical for mammary gland development and morphogenesis during pregnancy. PR-B also promotes uterine epithelial proliferation when not repressed by PR-A [12]. Furthermore, overexpression of PR-A led to endometrial hyperproliferation and infertility [19,20], revealing the importance of the relative PR-A/PR-B ratio to proper P4 responsiveness. Additionally, PGR has epithelial and stromal compartment-specific functions in the endometrium as revealed by ex vivo tissue-recombination experiments and in vivo epithelial-specific PGR knockout mice, while both epithelial and stromal PGR appear to be important for suppressing epithelial proliferation [21,22].

In response to P4 binding, PGR is capable of rapid, non-genomic action though interaction with c-Src kinase to induce the pro-proliferative extracellular signal-regulated kinase/mitogen-activated protein kinase (ERK/MAPK) and Protein Kinase B (AKT) pathways, important for peri-implantation stromal proliferation [12,23,24]. However, the canonical pathway for PGR’s impact on gene expression occurs through genomic activity after P4 binding and translocation to the nucleus [12]. Mechanistic studies of PGR action have been greatly aided by the identification of ligand-dependent PGR target genes in the mouse uterus through studies utilizing transcriptomic analysis of gene expression changes after P4 exposure in PRKO mice [25] and chromatin immunoprecipitation targeting PGR-bound gene regions [26]. One of the first PGR targets identified and known to be central to uterine function is the growth factor Indian hedgehog (IHH), which is induced in the epithelium and exerts paracrine effects on the stroma [27,28]. Uterine ablation of IHH in the mouse resulted in uterine phenotypes very similar to PGR knockouts [29]. Importantly, epithelial IHH induces stromal chicken ovalbumin upstream promoter-transcription factor II (COUPTFII) expression [27,29], which both inhibits E2-induced epithelial proliferation to allow implantation and induces bone morphogenetic protein 2 (BMP2) in the stroma to effect the decidualization response [30,31]. As shown in both mouse and human cells, BMP2 is critical for decidualization through induction of Wnt family member 4 (WNT4) [32,33], another ligand required for successful implantation and decidualization [34]. WNT4 and other Wnt family proteins canonically act through β-catenin activity [35], and β-catenin has also been implicated in uterine development, implantation, and decidualization [36]. In fact, a compartment-specific murine knockout of mesenchymal β-catenin showed that not only is stromal β-catenin required for decidualization, but it also indirectly opposes E2-induced epithelial proliferation [37].

Homeobox protein-A10 (HOXA10) is another PGR target in the endometrium, and HOXA10 knockout mice are infertile due to uterine defects that appear to be a result of lost stromal P4 responsiveness [38,39]. Interestingly, WNT4 expression is lost around the implantation site in HOXA10 mutant mice [40]. Adding to the complexity, heart and neural crest derivatives expressed 2 (HAND2), a stromal-expressed PGR target transcription factor, was also found to be required to mediate P4′s anti-proliferative action on the uterine epithelium but independently of IHH-COUPTFII signaling [41]. Rather, HAND2 inhibits stromal fibroblast growth factor (FGF) signaling [41], which otherwise induces epithelial proliferation through the ERK/MAPK and AKT pathways [42]. In addition, HAND2 appears to play a role in the decidualization process in both mouse and human stromal cells [43]. Outside of these more well-described pathways, many other P4 signaling mediators involved in uterine function and fertility have been described such as insulin-like growth factor binding protein 1 (IGFBP1) [44], CCAAT enhancer binding protein beta (C/EBPβ) [45,46], promyelocytic leukaemia zinc finger protein (PLZF) [47,48], mitogen-inducible gene 6 protein (MIG-6) [49,50], and cysteine rich secretory protein LCCL domain containing 2 (CRISPLD2) [51].

Though many of the important mediators and targets of uterine P4 signaling discussed above have been understood for many years, recent research, enabled by genome-wide transcriptome and cistrome analyses, has revealed new insight on P4 signaling regulators and modifiers [14]. Forkhead box O1 (FOXO1) was identified as a cell fate-regulating transcription factor involved in endometrial stromal decidualization partly through interaction with P4 signaling [52]. Further study revealed transcriptional cross-talk and greater than 75% overlap in genome binding occupancy between FOXO1 and PGR in in vitro human endometrial stromal cell decidualization, particularly in the regulation of Wnt signaling and other factors such as IGFBP1 [53,54]. Unexpectedly, a more recent in vivo mouse study found that rather than primarily functioning in decidualization, FOXO1 regulates epithelial integrity through regulation of PGR in vivo [55]. Indeed, conditional ablation of *Foxo1* in the uterus resulted in infertility primarily due to retention of epithelial integrity during the implantation window that prevented embryo invasion [55]. Transcriptomics and expression profiling further revealed a temporally and spatially controlled mutual regulation between PGR and FOXO1 in the uterine epithelium during the window of receptivity that was validated in human endometrial samples [55].

FK506 binding protein prolyl isomerase 4 (FKBP52) is a P4 signaling regulator from the FK506 binding family of immunophilins that was first found to interact with and promote PGR activity in vitro [56]. Targeted knockout of the *Fkbp52* gene in mice resulted in implantation failure resulting from attenuated P4-responsiveness due to a decrease in the binding of PGR by P4 [57]. Moreover, later findings revealed a strain-specific functional importance for FKBP52 mediating P4 responsiveness, highlighting the importance of genetics in its function [58]. In vitro decidualization experiments in human endometrial stromal cells confirmed a role for FKBP52 in decidualization and revealed HOXA10 as a regulator of FKBP52 in this process [59].

Signal transducer and activator of transcription 3 (STAT3) is a mediator of leukemia inhibitory factor (LIF) signaling [60] which will be discussed in more detail hereafter. The first clue to the importance of STAT3 in uterine function resulted from mouse implantation failure after pharmacological inhibition of STAT3 activation [61], and this result was later confirmed by the use of conditional gene knockouts that showed a decidualization defect, increased E2 signaling, and decreased P4 signaling [62,63]. More detailed analysis revealed that STAT3 directly interacts with PR-A, indicating a direct role for STAT3-PGR crosstalk in early pregnancy establishment [63].

GATA binding protein 2 (GATA2), a zinc finger family transcription factor, was originally identified as a PGR target in the mouse uterus via microarray analysis [25] and later confirmed to be expressed concomitantly with PGR in the uterine epithelium at temporally and spatially critical periods during pregnancy [64]. A follow-up study in which *Gata2* was conditionally ablated in the mouse uterus followed by genome-wide expression profiling and chromatin immunoprecipitation analysis revealed a large-scale regulatory role for GATA2 in PGR expression and downstream signaling [65]. *Gata2* uterine knockout mice were infertile due to implantation and decidualization defects, and further analysis showed that PGR protein and mRNA expression was dramatically reduced by *Gata2* attenuation [65]. Remarkably, 97% of P4-responsive genes failed to be induced without the presence of GATA2 as shown by microarray analysis [65]. Finally, cistrome analysis revealed that GATA2 both directly binds near the PGR promoter and shares occupancy with PGR at 50% of P4-responsive genes, and co-regulatory activity was confirmed with a luciferase reporter assay at *IHH* and sex determining region Y box 17 (*SOX17*) [65]. These results in the mouse were confirmed in the human by the finding of a correlation between GATA2 and PGR activity consistent with the mouse findings as well as a PGR-GATA2-SOX17 regulatory network governing female fertility [65].

SOX17 is a transcription factor identified as a PGR target by chromatin immunoprecipitation followed by massively parallel DNA sequencing (ChIP-seq) [26]. It was later found to be important in implantation, gland development, and gland function in the mouse uterus through experiments utilizing a knockout of one *Sox17* allele [66] and conditional knockouts of *Sox17* in PGR-positive cells and uterine epithelial cells [67]. More detailed study revealed that SOX17 controls epithelial proliferation and differentiation by regulating PGR signaling via the IHH pathway [68]. Furthermore, ChIP-seq analysis showed a remarkable overlap between SOX17, PGR, and GATA2-bound regions, and SOX17 was shown to induce IHH through direct binding of an enhancer 19 kb upstream to the *Ihh* gene [68]. Additionally, both the SOX17 expression pattern and a significant correlation with IHH expression were validated in human endometrial samples [68]. A further interesting note from this study was the high degree of correlation between the SOX17-regulated transcriptome and the (AT-rich interaction domain 1A (ARID1A)-regulated transcriptome in the mouse uterus at GD 3.5 along with the reduction of ARID1A expression in the SOX17-deleted uterus [68]. ARID1A is chromatin remodeling factor important for endometrial function that we will discuss in more detail hereafter [69].

### 2.2. Estrogen Receptors and Estrogen Signaling

P4 signaling in the endometrium cannot be considered on its own without also discussing the counteracting and sometimes cooperating action of E2 signaling. E2′s action in the endometrium is primarily enacted through the binding of its cognate nuclear receptors, estrogen receptor 1 (ESR1/ERα) and estrogen receptor 2 (ESR2/ERβ), which unlike PR-A and PR-B are transcribed from separate genes [2,4,5,70,71]. In addition to its classical genomic activity, ESR1 can also induce rapid non-genomic signaling through the ERK/MAPK pathway [72]. Specifically, ESR1 has been shown to promote proliferation through this pathway in a human epithelial cell line [73], and further evidence from mice indicates that ESR1 can successfully carry out its effects on endometrial epithelial proliferation independent of classical genomic signaling [74], suggesting a role for non-classical ESR1 activity in epithelial proliferation. Additional research has also shown a need for the ERK/MAPK pathway in endometrial stromal decidualization [75]. Much of the current understanding of uterine ESR1 and ESR2 was learned through a variety of genetically engineered mice as well as in vitro cell culture experiments. ESR2 knockout females show no apparent uterine defect and are subfertile only due to ovulation inefficiency with no difference in uterine E2-responsiveness [76,77]. However, there is some controversy because one study found competing evidence showing that the ESR2-null uterine epithelium is hyper-responsive to E2 treatment [78]. The first ESR1 knockout mouse was created using gene disruption in embryonic stem cells, and the resulting females were unresponsive to E2 and infertile with an ovarian defect and hypoplastic uteri [79] as well as depressed PGR expression [80]. Embryo transfer experiments showed that even with a healthy embryo and proper hormonal stimulation, uteri lacking ESR1 are not competent for implantation [81]. In spite of early reports to the contrary, likely resulting from incomplete deletion of ESR1 [70,80,81,82], ESR1 is also required in the mouse for a normal decidualization response to artificial stimulation [83]. Epithelial-specific ESR1 ablation resulted in the surprising finding that E2-induced epithelial proliferation occurs independently of epithelial ESR1, supporting previous findings from tissue recombination experiments [84,85]. Surprisingly, it is stromal ESR1 that controls E2-induced epithelial proliferation through stromal-epithelial crosstalk [85,86]. On the other hand, both epithelial and stromal ESR1 are necessary for a complete decidualization response to artificial stimulus [83,86].

The classic role for E2 in upregulating epithelial proliferation is mediated in part by insulin-like growth factor 1 (IGF1) downstream of ESR1 in the stroma [87,88]. Mechanistically, ESR1 induces IGF1 expression by interacting with a superenhancer distal from the IGF1 transcription start site [89,90]. It has been proposed that when IGF1 is expressed and secreted by the stroma, it binds its receptor IGF1R in the epithelium and induces the phosphoinositide 3-kinase (PI3K)/AKT pathway leading to proliferation [87,91,92]. However, it was recently shown that disrupting E2′s induction of IGF1 is not sufficient to the impair the E2-induced uterine growth response [89], so other mediators must be important as well. One family of such potential paracrine mediators is the FGF family, the members of which, as we mentioned earlier in our discussion of HAND2, induce the proliferation-associated ERK/MAPK and AKT pathways [41,42]. At least one FGF family member, FGF-9, is induced by E2 in the endometrial stroma [93]. In addition to its regulation by PGR in uterine stromal cells for decidualization, murine gene knockout experiments have shown that C/EBPβ is also an E2 target in both the endometrial epithelium and stroma that is critical for proliferation based on its activity regulating cyclin-dependent kinases in the Gap 2 (G2) to mitotic (M) phase cell cycle transition [45,46]. Finally, Mucin 1 (MUC1) is an E2 target in the uterine epithelium that is secreted to create a barrier to embryo attachment [94] until it is downregulated by P4 signaling through the IHH-COUPTFII pathway [29,30,31].

In addition to its activity inducing epithelial proliferation, the other critical role for E2 in the endometrium is the induction of LIF, an interleukin-6 family cytokine, in the glandular epithelium by the nidatory E2 spike [1,95]. Maternal LIF expression is absolutely required for successful implantation and decidualization in mice [96,97], and administration of LIF can replace the requirement of nidatory E2 for preparing a receptive uterus [97]. LIF induces downstream signaling in the luminal epithelium by binding its receptor (LIFR), which associates with glycoprotein 130 (gp130) and activates STAT3 through phosphorylation by Janus kinases (JAKs) [60,98]. As discussed previously in this review, activated phospho-STAT3 (pSTAT3) interacts with PGR signaling to promote implantation success and decidualization [63]. In addition, LIF action on the luminal epithelium regulates several important signaling pathways, some of which have been discussed here such as IGF1 signaling, Wnt/β-catenin signaling, FGF signaling, and ERK-MAPK signaling [95]. One mechanism of LIF action downstream of ESR1 was recently elucidated in which LIF acts through ERK1/2 to activate the IHH-COUPTFII pathway necessary for decidualization [83], revealing an additional layer of complexity in E2-P4 signaling crosstalk. Furthermore, the transcription factor early growth response 1 (EGR1) has been revealed as a regulator of implantation and decidualization induced by E2 through both the LIF-STAT3 and ERK1/2 pathways [99,100]. *Egr1* knockout mouse studies and human endometrial stromal cell in vitro decidualization experiments have established EGR1 as critical for endometrial receptivity through the regulation of epithelial PGR signaling [101], c-Kit expression [102], WNT4 expression [99], and many other cell-proliferation-related targets [103].

### 2.3. Nuclear Receptor Coregulators in the Regulation of Progesterone and Estrogen Signaling

Before turning to a focused discussion of P4 and E2 signaling dysregulation in endometriosis, the roles of nuclear receptor coregulators in steroid hormone signaling regulation must be briefly considered. In general, nuclear receptor coregulators form large complexes to modify chromatin structure and regulate large-scale gene transcription programs [104]. A family of regulatory proteins aptly named steroid receptor coactivators (SRCs), composed of SRC-1, SRC-2, and SRC-3, is critical to the regulation of PGR and ESR1 action in the female reproductive tract [105]. In the endometrium, SRC-1 and SRC-2 appear to be the most functionally relevant for normal functionality based on studies utilizing knockout mice [106,107,108,109,110,111]. This is supported by the fact that SRC-1 and SRC-2 are expressed more highly than SRC-3 in the human endometrium [112] although SRC-3 upregulation has been linked to endometrial cancer [113,114]. SRC-1 knockout mice are fertile; however, SRC-1 is necessary for full decidualization and P4-responsiveness in the uterus [106,109,111]. Intriguingly, SRC-1 appears to downregulate PGR target genes in the endometrial epithelium but upregulate them in the stroma [111]. SRC-2 is even more critical for murine uterine function. Uterine ablation of *SRC-2* resulted in complete female infertility due to implantation failure and a partial loss of decidualization which was completely lost with the concomitant ablation of *SRC-1* [107,110]. Microarray analysis further revealed that SRC-2 is necessary for P4 regulation of Wnt signaling, BMP2 signaling, and ESR1 signaling [110]. The requirement of SRC-2 for decidualization was also confirmed in in vitro decidualization of human endometrial stromal cells [115], and transcriptomic analysis revealed that 50% of SRC-2-regulated genes are also regulated by PGR [116], supporting the close relationship of these factors in transcriptional regulation of the decidualization process.

ARID1A, a SWItch/sucrose non-fermentable (SWI/SNF) chromatin remodeling complex protein, was recently found to be critical for endometrial function during early pregnancy after conditional deletion in the mouse uterus resulted in infertility due to implantation and decidualization defects [69]. *Arid1a* ablation also resulted in increased epithelial proliferation concurrent with increased epithelial E2 signaling and decreased epithelial PGR and P4 signaling [69]. Transcriptomic analysis indicated a role for ARID1A in repressing cell cycle related genes, and further experiments revealed that ARID1A complexes with PGR, specifically PR-A, to inhibit proliferation through the upregulation of Kruppel-like factor 15 (KLF15) [69] and to maintain an endometrium receptive to implantation.

Recent research has highlighted the importance of other epigenetic regulators in addition to ARID1A in endometrial P4 signaling. Enhancer of zeste homolog 2 (EZH2), a polycomb-repressive complex subunit that catalyzes histone 3 lysine 27 trimethylation and leads to gene silencing, was found to be involved in the epigenetic reprogramming required for decidualization [117]. Results from in vitro decidualization experiments indicated a role for EZH2 downregulation in decidualizing cells in response to progestin treatment. EZH2 is also upregulated in the endometrial epithelium by E2 in conjunction with increased epithelial proliferation, whereas P4 inhibits this effect [118]. Moreover, uterine deletion of *Ezh2* in the mouse compromised fertility [118]. Another epigenetic regulator, histone deacetylase 3 (HDAC3), functions by modifying histone acetylation, and this chromatin regulator was recently shown to be critical for implantation and decidualization in the mouse uterus and in vitro decidualization of human stromal cells [119]. Furthermore, uterine *Hdac3* knockout mice exhibited decreased PGR and PGR target gene expression in the stroma, indicating a role for HDAC3 in the P4-responsiveness required for stromal decidualization [119].

## 3. Dysregulation of Progesterone and Estrogen Signaling in Endometriosis

As the work reviewed in the previous sections has demonstrated, tightly regulated signaling pathways governed by P4 and E2 in a stromal and epithelial compartment-specific manner are key to maintaining endometrial homeostasis and supporting female fertility. Dysregulation of steroid hormone signaling is common in many uterine pathologies such as endometriosis, infertility, endometrial cancer, uterine leiomyoma, and recurrent pregnancy loss [12]. For the remainder of this review, we will focus on the molecular pathophysiology and treatment of endometriosis with particular focus on recent findings that shed light on the contribution of P4 and E2 signaling dysregulation to the infertility and pelvic pain women with this disease often experience.

Endometriosis is classically defined as the presence of endometrium-like tissue located outside the uterine cavity [8]. However, it is also important to understand this disease as a benign, heterogeneous, E2-dependent, and P4-resistant inflammatory condition that mainly affects the peritoneal cavity and ovary close to the uterus but has also been reported in distal organs such as the lungs and brain [9]. The prevalence of endometriosis is difficult to establish with certainty due to the requirement of surgical visualization of lesions for definitive diagnosis, but it is generally accepted that it occurs in about 1 in 10 women of reproductive age [8,9,120]. Several theories exist attempting to explain endometriosis pathogenesis such as peritoneal metaplasia or differentiation of circulating cells, but the most widely accepted explanation is the retrograde flow of menstrual tissue through the fallopian tubes [8,9]. Here, we will discuss evidence for the dysregulation of E2 and P4 signaling pathways in both endometriotic lesions (Table 1) and the endometriosis-affected eutopic endometrium that lead to P4 resistance and E2 dominance. These imbalances may explain the increased ability of lesions to grow outside the uterus and cause pain and the decreased ability of the uterus itself to support successful pregnancy establishment.

### 3.1. Progesterone Resistance

When endometrial tissue fails to respond properly to P4 exposure, this is termed P4 resistance, and it manifests itself in endometriosis as failed induction of PGR activation, or P4 target gene transcription in the presence of bioavailable P4 [6,7]. Under this definition, P4 resistance has been well-established in both the endometriotic lesions and eutopic endometrium of women with endometriosis [121,122,143,144]. Loss of P4-responsiveness can have serious consequences in both cases since P4 signaling is required to counteract E2-induced proliferation and to promote decidualization [7], which implies that P4 resistance may lead to both increased lesion growth and a non-receptive endometrium.

One potential molecular cause of P4 resistance is a loss or alteration of PGR expression, which has been documented in endometriotic lesions [121,122,123,124,125,126,127,128,129,130] and eutopic endometrium from women with endometriosis [124,130,145,146,147,148]. Further study has confirmed direct correlations between PGR loss with loss of P4-responsiveness in both lesions [149] and cells from the endometrium of women with endometriosis [150]. However, the contribution of PGR loss to the P4 resistance observed in endometriosis is controversial due to a few studies finding no significant difference in PGR levels in eutopic endometrium from women with endometriosis [123,151] or lesions [152]. These discrepancies are likely due to differences in experimental methods, lesion types and cell types analyzed, and resolution of PGR isoforms. For example, the two studies cited here finding no difference of PGR expression in the endometrium of women with endometriosis did not distinguish between PGR isoforms [123,151], and one did not distinguish between cell compartments either [151]. The study finding no difference of PGR expression in lesions looked specifically at rectosigmoid endometriosis lesions [152], whereas other studies found differences in PGR levels when analyzing mainly ovarian or peritoneal lesions [121,122,123,124,125,126,127,128,129,130]. Studies that distinguished between PR-A and PR-B tended to find a decrease of PR-B in endometriosis lesions [121,122,125,127] or endometrium [145,147,148], whereas reports of PR-A were mixed [121,122,129,130,145,146,147]. Furthermore, there is direct evidence to support promoter hypermethylation [127,153] and microRNA dysregulation [148,154] as potential mechanisms for PR-B loss in endometriosis. These findings support the importance of proper PR-A/ PR-B ratio in endometrial function and implicate an imbalance of PGR isoforms in the pathophysiology of endometriosis.

In addition to dysregulated PGR expression, alterations in PGR signaling mediators and regulators also contribute to P4 resistance [6]. Due to the importance of the PGR-induced IHH-COUPTFII-WNT4 pathway in regulating epithelial proliferation and decidualization during early pregnancy as discussed above, these molecules are of great interest in the context of P4 resistance in endometriosis. In a histological comparison of IHH expression in endometrial biopsy samples from women with endometriosis and healthy controls, IHH expression was decreased in secretory phase endometrium from endometriosis patients [155]. Correspondingly, later studies found COUPTFII [131] and WNT4 [132] expression levels decreased in both endometrial samples from women with endometriosis and endometriotic lesions. These findings identified a major pathway downstream of P4 signaling that is disrupted in women with endometriosis and may lead to endometrial non-receptivity in these patients by interfering with regulation of uterine epithelial proliferation and stromal decidualization. In endometriotic lesions, the PGR target HAND2 was also found decreased along with an increase in FGF signaling, which it normally controls [133]. This is another molecular consequence of P4 resistance that may lead to the increased invasiveness of endometriotic tissue [133]. Further confirming the comprehensive disruption of P4 signaling in endometriosis, HOXA10 [156], IGFBP1 [134], PLZF [144], MIG-6 [144], and CRISPLD2 [51], all PGR targets implicated in endometrial function based on mouse studies, have been shown to be dysregulated in endometriosis patients. These findings once again reinforce the idea that loss of P4 signaling in endometriosis disrupts the fine-tuned regulation of the endometrium necessary to maintain normal uterine function and fertility.

Though dysregulation of PGR target genes displays the consequences of P4 resistance in endometriosis, dysregulation of PGR signaling regulators may help explain the cause of P4 resistance. The expression of the pioneer transcription factor FOXO1 is reduced in both the endometrium [144,157] and stromal cells from lesions [122] of women with endometriosis. Given the requirement of FOXO1 for proper stromal cell decidualization and regulation of endometrial epithelial integrity along with the overlapping binding regions and in vivo regulation of PGR [53,54,55], the loss of FOXO1 in endometriosis could be partially responsible for the dysregulation of both PGR expression and downstream signaling. However, since FOXO1 is also regulated by PGR, it is difficult to conclude which molecule becomes dysregulated first in endometriosis based on the current literature. Another molecule with potential implications for P4 resistance in endometriosis is Notch homolog 1 (NOTCH1). NOTCH1 and other Notch signaling molecules have been found decreased in endometrium from women and baboons with endometriosis, and silencing of NOTCH1 impaired decidualization in isolated human endometrial stromal cells potentially by downregulation of FOXO1 [157], reminiscent of P4 resistance in endometriosis. Interestingly, aberrant NOTCH1 signaling has also been shown in endometriotic lesions, but in this case increased NOTCH1 activation correlated with reduced PGR expression [126]. In vitro reduction of NOTCH1 signaling restored PGR and P4-responsiveness, revealing a direct relationship between Notch signaling regulation and the maintenance of proper P4-responsiveness, both of which are disrupted in endometriosis [126].

Disruption of PGR signaling in endometriosis could also be caused by dysregulation of steroid receptor chaperone proteins like FKBP52. FKBP52 expression has been found decreased in both the endometrium and lesions of women with endometriosis [59,135], and the endometrial FKBP52 decrease alongside PGR decrease was confirmed to be due to endometriosis pathology in a non-human primate model of endometriosis [158]. Furthermore, endometriosis model mice lose FKBP52 expression in their lesions, and conversely, deletion of *Fkbp52* increased lesion growth [135]. HOXA10 may also be involved in this process since its expression is reduced in endometriosis [156], and in vitro experiments implicated it in the regulation of FKBP52 [59]. Evidence from both baboon and human endometriosis also implicates increased microRNA (miR)-29c expression as a potential mechanistic cause for FKBP52 loss [159].

STAT3 is another PGR regulator discussed earlier in this review with an important function in fertility [63]. Given its interaction with PGR during early pregnancy establishment, one might have hypothesized STAT3 activation would be reduced in endometriosis due to the context of P4 resistance, however pSTAT3 is aberrantly increased in the endometrium of both women and non-human primates with endometriosis [160]. This is likely explained by increased levels of interleukin 6 (IL-6) [161], which can activate STAT3 [162]. Abnormal STAT3 activity is associated with increased cell proliferation [163] which may occur through up-regulation of hypoxia-inducible factor 1-alpha (HIF1A) in the endometrium [160], illustrating its pleiotropic roles. Thus, while loss of STAT3 compromises uterine function, aberrant activation is associated with endometriosis, indicating the need for tight regulation of STAT3 in conjunction with PGR signaling. One potential mechanism suggested for increased STAT3 activation in endometriosis is down-regulation of protein inhibitor of activated STAT3 (PIAS3) which has been observed in women and non-human primates with endometriosis [164]. One effect of STAT3 overexpression in endometriosis appears to be the up-regulation of the oncogenic gene repressor B cell CLL/lymphoma 6 (BCL6), a known target of STAT3 [165] and shown to be increased in the secretory phase endometrium of women with endometriosis [166]. Furthermore, sirtuin 1 (SIRT1), a transcriptional regulator associated with both oncogenic and tumor-suppressor roles, binds and co-localizes with BCL6 and is also up-regulated in endometrium from women and non-human primates with endometriosis, significantly correlating with BCL6 expression levels [167]. Further experiments in mice and cell culture showed that increased BCL6 and SIRT1 expression caused reduced P4 signaling through the IHH pathway, specifically by binding the gene promoter of IHH pathway protein glioma-associated oncogene homolog 1 (GLI1) to repress its transcription [167]. In turn, reduced expression of GLI1 was shown in the endometrium of women with endometriosis, confirming a mechanistic role for STAT3, BCL6, and SIRT1 overexpression in the P4 resistance of endometriosis [167].

Earlier in this review we discussed the importance of the large-scale gene regulatory role for GATA2 and SOX17 in P4 signaling of the endometrium. In endometriosis, there appears to be a switch from a GATA2 driven P4-responsive state to a GATA6-driven P4-resistant state based on CpG methylation patterns [136]. Moreover, SOX17 expression is reduced in women with endometriosis, correlating with a drop in IHH expression, which it normally regulates by binding a distal enhancer to promote endometrial receptivity in the healthy endometrium [68]. In addition, ARID1A, a chromatin remodeling complex protein potentially regulated by SOX17, is decreased in endometrium from endometriosis patients [69]. Evidence showing direct binding of ARID1A to PR-A as well as loss of P4 signaling in mice with conditional ablation of *Arid1a* in the uterus implicates the decrease of ARID1A in endometriosis in the P4 resistance phenotype as well [69]. Expression of HDAC3, another epigenetic regulator, was also found decreased in endometrium from women with endometriosis as well as non-human primate and mouse models of endometriosis [119]. Further mechanistic study linked loss of HDAC3 to loss of P4 signaling, revealing yet another P4 signaling regulator implicated in the P4 resistance of endometriosis [119].

### 3.2. Estrogen Dominance and Inflammation

Concurrent with P4 resistance, endometriosis development and progression is driven by the upregulation of E2-induced cell proliferation and inflammation, which can both promote lesion growth and compromise endometrial receptivity [9,168,169]. Local E2 levels are increased in endometriosis due to upregulation of E2-producing p450 aromatase expression [170] and reduction of 17β-hydroxysteroid dehydrogenase type 2 (17βHSD2), which is normally induced by P4 to convert E2 to the less potent estrone but is decreased in P4-resistant conditions [171].

Since E2′s effects are primarily enacted through ESR1 and ESR2, their expression levels are important in the assessment of E2 action in endometriosis. ESR1 levels are reportedly increased in the secretory phase endometrium of women with endometriosis compared to controls [172,173], which may lead to increased estrogenic activity and proliferation, compromising normal uterine function. ESR2 expression is unchanged in eutopic endometrium from women with endometriosis [173] although one study reported increased ESR2/ESR1 ratio in endometriosis-affected endometrium [174]. The role of ESR2 in normal uterine physiology is not clear since ESR2 knockout mice have been reported to have no overt uterine defect [76,77]; however, one study implicated ESR2 in control of proliferation through epidermal growth factor (EGF) signaling [78].

In contrast, the majority of the evidence indicates that endometriotic lesions upregulate ESR2 and downregulate ESR1, although reports are mixed [136,137,138,139,140,142,175]. Discrepancies are likely due to the lesion type being studied since the majority of studies analyzed only ovarian lesions [136,137,138,139], but those including peritoneal lesions contrastingly showed relative increases in ESR1 [140,175]. Mechanistically, there is evidence to support changes in promoter methylation as a cause for the increase in the ESR2/ESR1 ratio in endometriotic cells, since regions of the *ESR1* promoter become hypermethylated, leading to decreased expression [136,137], whereas a CpG island in the *ESR2* promoter becomes hypomethylated, leading to increased expression [137]. Since E2 action through ESR1 upregulates PGR expression, the loss of ESR1 in lesions has been suggested as a possible explanation for the loss of PGR [9]. These mechanistic insights support the conclusion that the ESR2/ESR1 ratio increases in endometriotic lesions.

The increase in ESR2 levels in lesions may be responsible for increased lesion survival and inflammation because E2 can act through ESR2 to induce the cyclooxygenase-II (COX-2)-prostaglandin E2 (PGE2) feedback loop [176], which is well known to increase the inflammation and pathology of endometriosis [177]. E2 also induces ESR2 to bind the Ras-like, estrogen-regulated, growth inhibitor (RERG) promoter, inducing its expression [178]. In cooperation with PGE2, RERG was shown to translocate to the nucleus and induce cell proliferation, providing further evidence for the potential mechanism of E2-induced proliferation in endometriotic lesions [178]. Another study identified the E2-induced, proliferation-related proteins Myc proto-oncogene protein (c-myc), cyclin D1 (CCND1), and growth-regulating estrogen receptor-binding 1 (GREB1) as increased in expression alongside ESR2 in lesions [140], providing further clues to the mechanism of E2-dependent lesion growth. Additionally, FGF-9 is a cell growth-inducing factor shown to be induced by E2 and upregulated in endometriotic lesions [141], likely in part due to the loss of P4-induced HAND2 which would normally suppress it [41]. ESR2 upregulation in endometriotic lesions was reproduced in a mouse model of endometriosis, where its activity was shown to drive lesion growth and be an effective target for the inhibition of lesion growth [179]. Mechanistically, ESR2 apparently interacts with cytoplasmic inflammatory factors to inhibit apoptosis and promote the invasiveness of lesions [179].

Intriguingly, there is also evidence to implicate immune cell responsiveness to E2 in endometriosis. A growing body of evidence has implicated immune system dysregulation in endometriotic lesion growth, one aspect of which is elevated macrophage populations [180]. Peritoneal fluid macrophages from women with endometriosis were shown to upregulate the expression of ESR1 and ESR2, and the expression of ESRs correlated with an increase in inflammatory cytokines [181]. Further experiments in a mouse model of endometriosis showed that E2 treatment caused an increase in the macrophages present in lesions as well as the expression of macrophage migration factors [182]. In that study, ESR2 was the predominant E2 receptor expressed in macrophages from both women with endometriosis and endometriosis model mice [182]. Thus, E2 appears to directly cause an increased inflammatory response through ESR2 in addition to enhancing endometriotic cell proliferation in endometriosis.

In addition to the targets of E2 and ESRs that induce cell proliferation and inflammation in endometriosis, it is also important to consider the potential effects of SRCs on ESRs in endometriosis. Expression profiling of SRCs in endometriotic lesions identified SRC-1 as the predominant SRC in endometriosis [183]. Although one study found that SRC-1 expression was decreased in the epithelium of proliferative phase endometriotic lesions [184], additional research initiated in endometriosis model mice and validated in human endometriosis revealed that in spite of a decrease in total SRC-1, levels of a truncated form were increased [142]. Furthermore, this new isoform of SRC-1 was shown in vitro to decrease tumor necrosis factor alpha (TNFα)-mediated apoptosis in endometriotic cells, leading to increased cell survival and invasion and mirroring the in vivo disease pathophysiology [142]. Additional experiments revealed interaction between this SRC-1 isoform and ESR2 in endometriosis that may mediate a synergistic role in promoting cell survival [179]. Indeed, disruption of the SRC-1 isoform-ESR2 access with inhibitors suppressed endometriotic cell growth in isolated human cells and in a mouse model of endometriosis [179,185]. Taken together, these findings support an important role for SRC-1 isoform and ESR2 upregulation in the development and progression of endometriosis.

Although LIF expression is induced by E2 in the endometrium, and estrogenic activity is increased in endometriosis, LIF levels have been reported to be decreased in the glandular epithelium of women with this disease [186]. This could be due to increased inflammatory factors in endometriosis that can suppress LIF [187]. The decrease in LIF secretion from glands may also be due to intrinsic gland dysfunction in endometriosis. Specifically, the gland-specific transcription factor Forkhead box A2 (FOXA2) is required for LIF expression in mice [188], but it is decreased in endometrium from women with endometriosis [139,189]. Thus, though increased estrogenic activity promotes harmful inflammation and cell proliferation in endometriosis, it apparently fails to properly induce LIF expression.

## 4. Pathologies Related to Steroid Hormone Signaling Dysregulation in Endometriosis

### 4.1. Infertility

One of the major clinical pathologies associated with endometriosis is infertility [168,190,191,192,193,194]. Although up to 50% of women with endometriosis struggle with fertility problems, the causal link is unclear and controversial [190]. Several possible mechanisms have been proposed by which endometriosis may cause fertility defects including (1) anatomical distortions, (2) diminished ovarian reserve, (3) chronic inflammatory conditions, and (4) compromised endometrial receptivity [191,193]. Due to the well-studied involvement of P4 and E2 signaling in endometrial receptivity, we will focus our discussion of P4 and E2 dysregulation in endometriosis-related infertility on that topic (Figure 1).

Because of the integral involvement of P4 and E2 signaling pathways in early pregnancy establishment and their dysregulation in the endometriosis-affected endometrium that we have described above, it is intuitive to draw a conceptual link between the P4 resistance and E2 dominance of endometriosis and the endometrial non-receptivity associated with this disease. In addition to the broad conceptual link, several specific molecular pathways we have discussed are implicated in both female infertility and endometriosis. For example, total endometrial PGR expression and PR-A/PR-B expression ratio are critical for successful mammalian pregnancy as shown primarily in mice [16,17,18,19,20], but either PGR total expression or PR-A/PR-B ratios are dysregulated in the endometrium of many women with endometriosis [124,130,145,146,147,148]. In fact, a recent translational study showed that in women diagnosed and treated for endometriosis, PGR expression levels were higher in women with subsequent spontaneous pregnancies within one year versus those who did not successfully achieve pregnancy [195]. Additionally, the inhibitory action of PGR on ESR1 normally prevails in the endometrium during the window of receptivity, but women with endometriosis exhibit increased ESR1 through the mid-secretory phase, which encapsulates the implantation window in women [172,173].

The rise in ESR1 in endometriosis corresponds to a decrease in αv/β3 integrin [196], which is an adhesion molecule normally expressed in the endometrium during the receptive window and putatively involved in successful implantation [172]. HOXA10, a P4 target decreased in the endometrium of women with endometriosis [156] and required for fertility in mice [38,39], was also identified as a direct regulator of αv/β3 integrin expression [197]. Furthermore, non-human primates induced with endometriosis exhibit reduced HOXA10 and αv/β3 integrin expression [198]. In addition to regulating αv/β3 integrin, HOXA10 regulates FKBP52, a PGR regulator required for implantation and decidualization [57,59] and reduced in the endometrium of women and non-human primates with endometriosis [59,135,158].

Several other steroid hormone-regulated pathways we have discussed are both implicated in endometriosis and closely involved in pregnancy establishment. Proteins involved in regulation and mediation of the P4-responsive IHH pathway including GATA2, SOX17, IHH, COUPTFII, and WNT4 are required for successful implantation in mice [29,30,34,65,67] but are reduced in the endometrium of women with endometriosis [68,131,132,136,155], revealing a potential large-scale molecular connection between P4 resistance and fertility problems in endometriosis. The transcriptional regulators FOXO1, ARID1A, and HDAC3 are three additional factors associated with P4 signaling that are required for uterine receptivity in mice and down-regulated in the endometrium of women with endometriosis, further corroborating the association between the P4 resistance of endometriosis with infertility [55,69,119,144,157]. Finally, the E2-responsive cytokine LIF, required for fertility in mice and women [96,97], is both decreased generally in endometrium from women with endometriosis [186] and specifically correlated with failure to achieve pregnancy in women with the disease [199]. Taken together, the evidence of dysregulation in these pregnancy-associated pathways and molecules in endometriosis is a strong indicator of a causative relationship between endometriosis and endometrial non-receptivity related to P4 and E2 signaling dysregulation.

### 4.2. Pelvic Pain

In addition to infertility, it is commonly known that many women with endometriosis experience pelvic pain. Indeed, one study found 80% of women diagnosed with chronic pelvic pain to have endometriosis, firmly establishing the association [200]. Several mechanisms have been proposed for the pain of endometriosis including invasion of lesions into highly innervated regions, increased endometrial nerve density, increased neuroangiogenesis, neuroinflammation, and central and peripheral sensitization [201,202,203]. A comprehensive discussion of endometriosis pain is not our purpose here, but we will briefly mention several links that have been discovered between E2 signaling and the pain mechanisms of endometriosis. First, several factors involved in nerve growth and found upregulated in women with endometriosis [8] have been found to be regulated by E2, including nerve growth factor (NGF) [204], vascular endothelial growth factor (VEGF) [205], and brain-derived neurotrophic factor (BDNF) [206]. Additionally, hormonal therapies designed to combat the E2 dominance of endometriosis have been shown to decrease endometrial nerve fiber density in women with endometriosis, implying a role for E2 in increased innervation [207]. E2 has also been implicated in the neuroinflammation of endometriosis by increasing macrophage-nerve interactions in endometriotic lesions [182]. Furthermore, a recent study revealed a role for E2 in regulating the axonal guidance protein slit guidance ligand 3 (SLIT3) in the process of neuroangiogenesis in endometriotic lesions [208]. Finally, nociceptors are sensory nerve endings that transmit noxious stimuli to the central nervous system in the presence of potential damage, and transient receptor potential cation channel subfamily V member 1 (TRPV1), an ion channel protein associated with these neurons, has been found to be increased in endometriotic lesions of women with chronic pelvic pain [209] and to be responsive to E2 [210]. These mechanisms, among others, are potential avenues by which E2 elevation in endometriosis may worsen the pain associated with the disease.

## 5. Hormone Therapies for Endometriosis

Treatments for endometriosis that aim to alter E2 and P4 signaling are currently in use, such as combined oral contraceptives (COCs), progestins, gonadotropin-releasing hormone (GnRH) agonists, and aromatase inhibitors, and others are still under development, such as GnRH antagonists, selective estrogen receptor modulators (SERMs), and selective progesterone receptor modulators (SPRMs) [211,212,213] (Table 2). These treatments generally aim to treat the lesion growth itself and/or the pelvic pain of the disease. Currently, no medical treatments are available to treat the infertility associated with endometriosis because hormone therapies interfere with ovarian function [194], although some evidence indicates a possible benefit to timed treatments combined with surgery or assisted reproductive technologies [168].

Several medical treatments for endometriosis directly aim to reduce E2 production or action in order to mitigate E2 dominant conditions. GnRH agonists are normally second-line treatments that decrease hormone levels by downregulating the pituitary through negative feedback mechanisms [213]. In randomized controlled trials (RCTs), GnRH agonists have been shown to be effective in reducing endometriosis-related pain [214], but they may also have adverse effects such as bone mineral density loss due to a hypoestrogenic state, requiring hormone “add-back” for long term use [213]. GnRH antagonists are also currently under investigation for endometriosis treatment. Like GnRH agonists, they downregulate gonadotropins, but they do not cause flare-ups like GnRH agonists because they rapidly and directly compete for GnRH receptors [215]. A recent RCT showed one GnRH antagonist to be effective at reducing endometriosis pain but to have similar hypoestrogenic adverse effects as GnRH agonists [216]. Aromatase inhibitors such as anastrazole or letrozole are also in use for some endometriosis patients, although they are recommended only for women who do not respond to other available treatments due to severe side effects [217]. In a mouse model of endometriosis, aromatase inhibitors decreased lesion size by increasing apoptosis and diminishing VEGF and PGE2 levels [218]. Clinical trials have shown some success for aromatase inhibitors in reducing chronic pelvic pain, but significant adverse effects such as irregular bleeding and joint pain have been reported [219]. There is also a relatively new category of drugs under investigation aimed at targeted downregulation of E2 signaling termed SERMs, and these bind directly to ESRs in a tissue-specific manner [213]. These have been shown to reduce lesions through downregulation of ESR1 and cell proliferation in rat models of endometriosis [220,221], but a clinical trial in which treatment group endometriosis pain returned more quickly after surgery tempers enthusiasm presently and points to the need for further study before SERMs can be broadly used [222].

Other medical treatments for endometriosis primarily center on treating the dysregulation of P4 signaling in endometriosis. COCs consisting of a formulation of E2 and progestins that suppress ovarian steroid production are often used as a first-line therapy for chronic treatment of endometriosis pain due to their practical benefits and safety over long-term use [217]. COCs have shown efficacy in preventing endometriosis recurrence after surgical removal of lesions [223] as well as pain associated with the disease [224,225]. It has been suggested that progestin-only therapies may be a better choice since the inclusion of E2 could exacerbate estrogenic conditions in the context of P4 resistance [238]. Progestin-based therapies such as norethisterone acetate, levonorgestrel, and medroxyprogesterone acetate (MPA) are another first-line treatment choice for endometriosis pain [217]. These compounds cause decidualization and atrophy of both the eutopic endometrium and endometriotic tissue [212] and have proven to be effective at reducing endometriosis-related pain in clinical trials [226,227,228]. MPA was specifically shown to have equivalent efficacy to a GnRH agonist at reducing pain but without the adverse hypoestrogenic effects on bone density [227]. In fact, one study revealed that MPA can decrease ESR1 and ESR2 while increasing PR-A and PR-B in the endometrium women with endometriosis [229]. Another drug in this category, danazol, works by promoting a high androgen, low E2 environment [230]. Danazol has demonstrated efficacy in treating pain and reducing lesions in endometriosis, but significant androgenic side-effects occur [230]. A more recently developed progestin, dienogest (DNG), shows much promise. DNG has been shown to successfully reduce endometriosis-associated pelvic pain with limited adverse effects such as minor irregular bleeding [231,232]. Furthermore, DNG inhibits endometriotic stromal cell proliferation [233], increases the PR-B/PR-A ratio and decreases the ESR2/ESR1 ratio [234], and inhibits E2 production and aromatase expression [235].

While progestins are an effective treatment option for many women with endometriosis, P4 resistance renders many others unresponsive to progestin treatment [239]. This dilemma serves as a call for new treatment strategies, one of which may be SPRMs currently under investigation. These drugs interact directly with PGR to alter its downstream effects for the purpose of reducing proliferation and prostaglandin production [212]. Mifepristone trials have indicated its efficacy in endometriosis pain improvement and lesion reduction, although results are mixed [236,237]. One early report indicated asoprisnil, which has mixed P4 agonist/antagonist activity and endometrial selectivity, also succeeded in lowering endometriosis-related pain, but this trial was apparently cut short due to some women developing endometrial hyperplasia [212]. Clearly, further investigation must be carried out to assess the safety and efficacy of this class of molecules, but it represents a potential new avenue for women who do not respond to currently available therapies.

## 6. Conclusions

Endometrial homeostasis is clearly tightly interwoven with P4 and E2 signaling. Mutual cooperation and regulation of P4 and E2 pathways is absolutely critical for uterine function and fertility, and when dysregulation of these pathways occurs, pathologies like endometriosis and infertility are common. Recent work enabled by sensitive cistromic and transcriptomic analyses has particularly revealed critical details of P4 and E2 signaling regulators and modulators that are key to maintaining healthy gene regulatory networks in the endometrium. Deepening knowledge of the molecular underpinnings of normal P4 and E2 signaling in the endometrium juxtaposed with the dysregulation in endometriosis that results in E2 dominance and P4 resistance has begun to enable more targeted and individualized treatment options for women suffering the effects of endometriosis, and further study in this area has great potential to unlock treatment opportunities with more permanent efficacy and with fewer adverse effects than currently available therapies can provide.

## Figures and Tables

**Figure 1 ijms-20-03822-f001:**
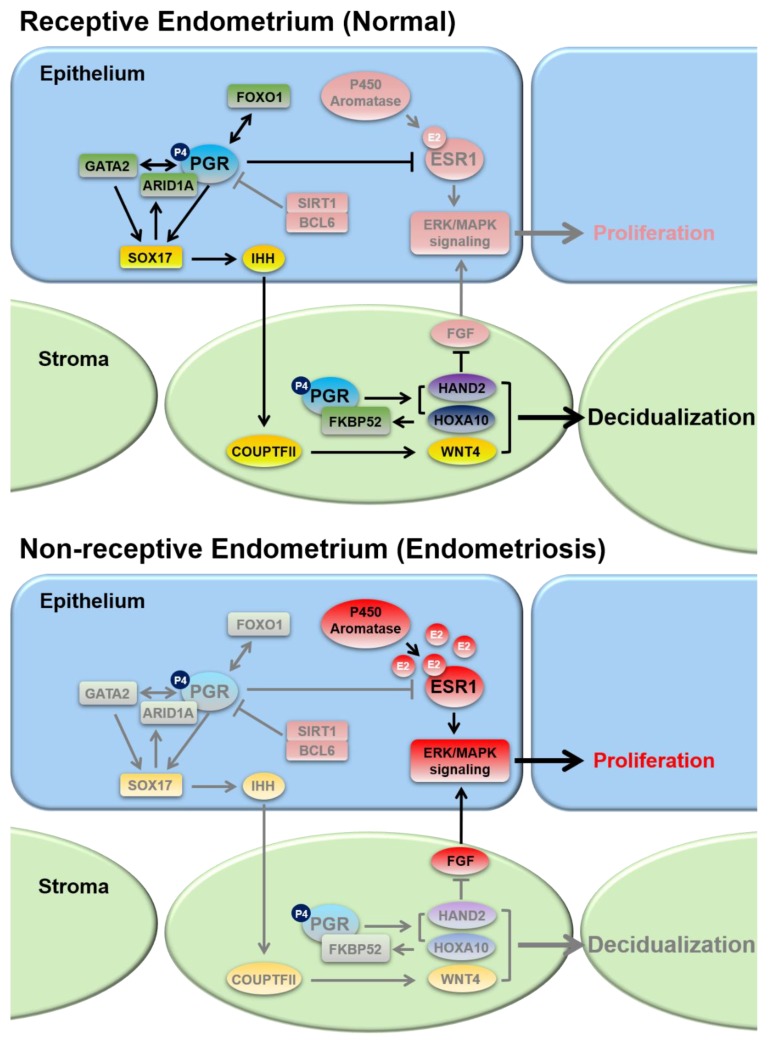
Schematic diagram illustrating the primary known signaling pathways and transcriptional regulators involved in P4 and E2 governance of endometrial epithelial-stromal crosstalk that are dysregulated in endometriosis. P4 resistance and E2 dominance in endometriosis results in epithelial proliferation and defective decidualization that can compromise endometrial function. Abbreviations: ARID1A, AT-rich interaction domain 1A; BCL6, B cell CLL/lymphoma 6; COUPTFII, chicken ovalbumin upstream promoter-transcription factor II; E2, estrogen; ERK, extracellular signal-regulated kinase; ESR1, estrogen receptor 1; FGF, fibroblast growth factor; FKBP52, FK506 binding protein prolyl isomerase 4; FOXO1, Forhead box O1; GATA2, GATA binding protein 2; HAND2, heart and neural crest derivatives expressed 2; HOXA10, homeobox protein-A10; IHH, Indian hedgehog; MAPK, mitogen-activated protein kinase; P4, progesterone; PGR, progesterone receptor; SIRT1, Sirtuin 1; SOX17, sex determining region Y box 17; WNT4, Wnt family member 4.

**Table 1 ijms-20-03822-t001:** P4 and E2 signaling factors dysregulated in endometriotic lesions.

	Molecule	Symbol	Function	Dysregulation	Reference
**P4 Signaling Factors**	Progesterone Receptor	PGR	Nuclear receptor	Decreased	[121,122,123,124,125,126,127,128,129,130]
	Chicken ovalbumin upstream promoter-transcription factor II	COUPTFII	Transcription factor	Decreased	[131]
	Wnt family member 4	WNT4	Secreted signaling protein	Decreased	[132]
	Heart and neural crest derivatives expressed 2	HAND2	Transcription factor	Decreased	[133]
	Insulin-like growth factor binding protein 1	IGFBP1	Circulating growth factor binding protein	Decreased	[134]
	Forkhead box O1	FOXO1	Transcription factor	Decreased	[122]
	FK506 binding protein prolyl isomerase 4	FKBP52	Immunophilin	Decreased	[135]
	GATA binding protein 2	GATA2	Transcription factor	Decreased	[136]
**E2 Signaling Factors**	Estrogen receptor 1	ESR1	Nuclear receptor	Decreased	[136,137,138,139]
	Estrogen receptor 2	ESR2	Nuclear receptor	Increased	[136,137,138,139]
	Myc proto-oncogene protein	c-MYC	Transcription factor	Increased	[140]
	Cyclin D1	CCND1	Cell cycle regulator	Increased	[140]
	Growth regulating estrogen receptor binding 1	GREB	Growth regulator	Increased	[140]
	Fibroblast growth factor 9	FGF-9	Secreted growth factor	Increased	[141]
	Steroid receptor coactivator-1	SRC-1	Transcriptional co-activator	Increased	[142]

**Table 2 ijms-20-03822-t002:** Hormone therapies for endometriosis.

	Treatment Type	Molecular Action	Therapeutic Effect	Reference
**Estrogen (E2) Signaling Modifiers**	Gonadotropin-releasing hormone (GnRH) agonists	Decrease E2 production through negative feedback	Reduce endometriosis-related pain	[213,214]
	GnRH antagonists	Decrease E2 production by competing for GnRH receptors	Reduce endometriosis-related pain	[215,216]
	Aromatase inhibitors	Decrease E2 production by inhibiting conversion of androgens to E2	Reduce endometriosis-related pain and lesion size	[217,218,219]
	Selective estrogen receptor modulators (SERMs)	Decrease estrogen receptor 1 (ESR1) action through direct inhibition	Reduce endometriotic lesions	[213,220,221,222]
**Progesterone (P4) Signaling Modifiers**	Combined oral contraceptives (COCs)	Suppress ovarian steroid production and supplement P4 levels	Reduce endometriosis-related pain and recurrence after surgery	[217,223,224,225]
	Progestins	Supplement P4 levels	Reduce endometriosis-related pain and lesions	[212,217,226,227,228,229,230,231,232,233,234,235]
	Selective progesterone receptor modulators (SPRMs)	Interact with progesterone receptor (PGR) to enhance downstream effects	Reduce endometriosis-related pain and lesions	[212,236,237]

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
