# Peer review of "Progesterone and Estrogen Signaling in the Endometrium: What Goes Wrong in Endometriosis?"

_ijms, 2019, doi:10.3390/ijms20153822_

Round 1
Reviewer 1 Report
The paper “Progesterone and Estrogen Signaling in the Endometrium: What Goes Wrong in Endometriosis?” by Marquardt and colleagues is an interesting review about the molecular mechanisms underlining the Estrogen and Progesteron function in endometrium in physiological condition and their dysregulation in endometriosis. The article is well written and cover the current state of the field exhaustively, thus in my opinion the paper is suitable for publication in “International Journal of Molecular Sciences” once addressed some points as reported below.
Page 1 Lines 36-40 The sentence “ The progesterone (P4) and estrogen (E2)-responsive signaling pathways integral for early pregnancy success which are primarily induced genomically through binding nuclear receptors—the progesterone receptor (PGR) for P4 and estrogen receptors (ESR1 and ESR2) for E2—are regulated in an epithelial and stromal compartment-specific manner (reviewed in 39 ref. [3-5]).” is not fully clear and it should be revised.
Page 4 Lines 42-44 Author state “In addition to its genomic activity, ESR1 can also induce rapid non-genomic signaling through the MAPK-ERK1/2 pathway (reviewed in ref. [72]), which is involved in endometrial stromal decidualization [73]. “, however as reported in the figure 1 it appears that E2 rapid effects on MAPK/ERK in epithelium are directly involved in the epithelium proliferation. Authors should further discuss this reported effect of E2 rapid non genomic signaling on epithelium proliferation in the text.
In the paragraph 3.2 authors report “In contrast, the majority of the evidence indicates that endometriotic lesions upregulate ESR2 and downregulate ESR1, although reports are mixed“ highlighting, from my point of view, a critical function of ESR2, and, in particular, its upregulation in the endometriosis. To fully cover this issue, authors should further discuss the role of ESR2 also in physiological endometrium condition and draw, for a complete understanding, how this receptor could be involved in endometriosis and in the associated inflammation by adding it in the figure 1 (or another image).
Author Response
We want to sincerely thank reviewer 1 for detailed review of this manuscript and for the insightful comments, which have helped improve the quality of the manuscript.
Point 1: Page 1 Lines 36-40 The sentence “The progesterone (P4) and estrogen (E2)-responsive signaling pathways integral for early pregnancy success which are primarily induced genomically through binding nuclear receptors—the progesterone receptor (PGR) for P4 and estrogen receptors (ESR1 and ESR2) for E2—are regulated in an epithelial and stromal compartment-specific manner (reviewed in 39 ref. [3-5]).” is not fully clear and it should be revised.
Response 1: This sentence has been revised for clarity. It now reads, “The progesterone (P4) and estrogen (E2)-responsive signaling pathways integral for early pregnancy success are primarily induced through their cognate nuclear receptors, the progesterone receptor (PGR) and estrogen receptors (ESR1 and ESR2), respectively. These pathways are regulated in an epithelial and stromal compartment-specific manner in the endometrium (reviewed in ref. [3-5]).”
Point 2: Page 4 Lines 42-44 Author state “In addition to its genomic activity, ESR1 can also induce rapid non-genomic signaling through the MAPK-ERK1/2 pathway (reviewed in ref. [72]), which is involved in endometrial stromal decidualization [73]. “, however as reported in the figure 1 it appears that E2 rapid effects on MAPK/ERK in epithelium are directly involved in the epithelium proliferation. Authors should further discuss this reported effect of E2 rapid non genomic signaling on epithelium proliferation in the text.
Response 2: Additional discussion of E2-induced MAPK/ERK effects on epithelial proliferation has been added on Page 4 lines 4-47.
Point 3: In the paragraph 3.2 authors report “In contrast, the majority of the evidence indicates that endometriotic lesions upregulate ESR2 and downregulate ESR1, although reports are mixed“ highlighting, from my point of view, a critical function of ESR2, and, in particular, its upregulation in the endometriosis. To fully cover this issue, authors should further discuss the role of ESR2 also in physiological endometrium condition and draw, for a complete understanding, how this receptor could be involved in endometriosis and in the associated inflammation by adding it in the figure 1 (or another image).
Response 3: Figure 1 illustrates the molecular pathways in receptive endometrium and non-receptive endometrium from endometriosis patients. Therefore, ESR2 has been added to Graphical Abstract to show the role of ESR2 in inflammation and cell proliferation. The role of ESR2 in the physiological endometrium has been added to this section on page 9 lines 39-41.
Reviewer 2 Report
First of all, I would like to point out that this article completely fits the Section “Molecular Pathology, Diagnostics, and Therapeutics, Molecular Pathophysiology of Uterine and Ovarian Diseases and Dysfunction”.
As indicated by the authors the review is focused on the description of progesterone and estrogen signaling in “healthy” endometrium and during endometriosis. Overall, the paper is clearly written and interesting to read. It should be specifically highlighted that a lot of very recent findings are discussed in the review. However, I need to mention that the study seems a bit descriptive, lacking authors’ own opinion. Nevertheless, to my mind such a review is worth publishing.
Author Response
We want to thank reviewer 2 for carefully reviewing this manuscript and for the positive comments and constructive criticism.